# A new STI in the city: MPOX in Barcelona. First outbreak (5/2022-5/2023) and subsequent resurgence

David Palma[1,2,3]*, Montserrat Guillaumes[1,4], Carles Pericas[1,4,5], Anna de Andrés[1], Raquel Prieto[1,6], Laia Álvarez-Bruned[1], Jesús Ospina[1,3], Paula Santiá[1,6], Patricia García de Olalla[1,3,4], Cristina Rius[1,3,4,6]

1 Agència de Salut Pública de Barcelona (ASPB), Barcelona, Spain, 2 Department of International Health, Care and Public Health Research Institute (CAPHRI), Maastricht University, Maastricht, the Netherland, 3 Consorcio de Investigación Biomédica en Red en Epidemiología y Salud Pública (CIBERESP), Madrid, Spain, 4 Institut de Recerca de l'Hospital de la Santa Creu i Sant Pau (IIB Sant Pau), Barcelona, Spain, 5 Departament de Medicina, Universitat de Barcelona, Barcelona, Spain, 6 Department of Experimental and Health Sciences, Faculty of Health and Life Sciences, Universitat Pompeu Fabra (UPF), Barcelona, Spain

* ext_dpalma@aspb.cat

**Data Availability Statement:** All files are available from the Zenodo database DOI: 10.5281/zenodo.13284254.

## Abstract

### Background

In May 2022, after the suspension of the mobility restrictions due to the COVID-19 pandemic, the first outbreak of MPOX virus, transmitted from human to human, was detected outside of Africa, affecting mostly sexually active men who have sex with men. Our aim is to report the first outbreak of MPOX in Barcelona city in the period from 5/2022 to 5/2023 and the subsequent surge of cases in 8/2023.

### Methods

We performed a descriptive study of all notified cases in city residents, obtained through epidemiological surveys. The analyses are presented for the hospitalized cases and the key population of men who have sex with men.

### Results

Of 2037 notified cases, 82.6% were confirmed. The cumulative incidence in the general population was 1.03 (95%CI 1.00–1.06) per 1000 inhabitants and 2.13 (2.01–2.17) per 1000 in men. Men were older than women, with a median age of 37 years (interquartile range 31–43). While 84.5% of men reported having sex with partners of the same gender, 70.9% of women only reported sex with partners of the opposite gender. Complications occurred in 4.1% of infected persons, 1.6% required hospitalization, and no deaths were registered. Georeferencing was highly associated with gay venues. Gay, bisexual and other men who have sex with men (GBMSM) accounted for most cases and severe cases, and were associated with attending public sex venues and not providing contact tracing information. Digital and printed prevention campaign materials were developed for GBMSM.

**Funding:** The author(s) received no specific funding for this work.

## Discussion

The 2022 MPOX outbreak posed a major challenge to surveillance and sexual health services worldwide. With new cases and re-infections on the rise, MPOX may become a regular infection to be incorporated in STI testing and management guidelines. Barcelona has some characteristics that may facilitate the occurrence and spread of emergencies related to sexual health.

## Background

The monkeypox virus (MPOX; family *Poxviridae*, genus *Orthopoxvirus*) is a large (200–250 nm), enveloped, double-stranded DNA virus [1]. The virus was first identified in 1958 in Copenhagen during two outbreaks among captive cynomolgus monkeys from Singapore [2]. Although cases are suspected to have occurred for thousands of years [3], it is likely that the eradication of smallpox revealed the continued occurrence of smallpox-like illnesses in rural areas of sub-Saharan Africa [3]. Since the initial description of the disease, the number of local outbreaks across Central and West Africa has increased, and it is now considered to be an endemic zoonosis [4].

Following the final recommendations of the Global Commission to eradicate smallpox [5], an active surveillance program for human MPOX was established in the DRC between 1981 and 1986, which may have impacted the increase in incidence over the last decades [6]. In July 2003, the first human outbreak outside Africa was reported in the USA, with 71 non-fatal cases, and was related to 800 specimens from six genera of African rodents imported from Ghana [7]. During this period, transmissibility between rodents, rabbits and dogs was demonstrated, although the probability of human-to-human transmission was low, as no cases were attributed to secondary transmission.

In 2018, the World Health Organization (WHO) identified MPOX as an emerging disease on their list of priority diseases and called for enhanced surveillance measures [8]. Further imported cases from Nigeria were reported in the UK [9], Israel [10], and Singapore [11]. In 2020, modeling studies suggested an MPOX $R_0$ between 1.10 and 2.40 in countries where exposure to *Orthopoxvirus* species is negligible, suggesting the imminent threat of an MPOX epidemic in scenarios involving imported human or animal cases [12]. In 2021, two cases occurring in persons returning from Nigeria were reported, one in Texas (July) and other in Washington (November). On May 6, 2022, a UK resident returning from Nigeria was confirmed as a case [13].

## Outbreak detection

On May 18, 2022, seven cases were reported in Spain, 14 in Portugal, and 13 in Canada. The next day, cases were confirmed in Belgium, Sweden, and Italy. Two days later, cases were reported in France, Germany, the Netherlands, the UK, and Australia, with two cases returning from Europe [13]. On July 23, with an accumulated 3,040 cases from 47 countries, the WHO Director-General declared the escalating global MPOX outbreak a Public Health Emergency of International Concern (PHEIC) [14], primarily affecting gay, bisexual and other men who have sex with men (GBMSM) [1].

On May 19, the first case was reported in the city of Barcelona, Spain, involving a 44-year-old male resident known to have attended a mass event during international travel in the past

21 days. The most probable mechanism of transmission was sexual contact with another men. On 21 May, the case was confirmed, accompanied by an increasing number of daily cases. We report the complete first outbreak in the city and the measures undertaken by the Epidemiology Department of the Public Health Agency of Barcelona, up to May 10, 2023, when, after six weeks without any notified, confirmed, or suspected cases, the outbreak was declared closed. We also briefly describe the onset of the second wave cases and lessons learned.

## Methods

### Study design

We conducted a descriptive study of the cumulative incidence from all notified MPOX cases during the outbreak among residents of the city of Barcelona between May 2022 and May 2023.

### Definitions of cases and close contacts

According to the guidelines of the Spanish Ministry of Health [15], MPOX was designated an urgently notifiable disease, requiring an epidemiological survey to be completed individually for each case. A suspected case was defined as one showing clinical criteria highly suggestive of MPOX infection [14–16], while other diagnoses were ruled out. A probable case was defined as one meeting both clinical and epidemiological criteria, such as having been in close contact with a confirmed or probable MPOX case, reporting high-risk sexual partners, or recent travel to endemic areas in Western or Central Africa. A case was confirmed when laboratory results were positive for specific MPOX or generic *Orthopoxvirus* in a polymerase chain reaction (PCR) test.

### Inclusion and exclusion criteria

All notified cases of people residing in Barcelona were included to chart the evolution of the outbreak. Subsequently, only confirmed cases, regardless of age, were included in the analysis. Suspected or probable cases that were not confirmed by laboratory results, as well as confirmed cases living outside Barcelona, were excluded from the analysis.

### Management of cases and close contacts

The Spanish protocol initially guided all actions taken by the Public Health Agency of Barcelona. An internal notification system was established to follow-up all cases and conduct a thorough assessment of sexual partners (Fig 1).

All sexual partners of confirmed cases were informed of their close contact with someone diagnosed with MPOX and were provided with instructions to identify and monitor potential symptoms for at least three weeks. They were advised to contact health professionals if any symptoms of infection developed. Follow-up of sexual partners was prioritized for individuals from vulnerable groups, including pregnant women, immunocompromised individuals, and minors. The contact tracing study was conducted prospectively from the first case until the capacity of the epidemiology surveillance workers was reached. Following this, the study was extended retrospectively until the end of February 2023 with the assistance of community health workers proficient in a variety of languages. In July, the Catalan Health Department released a new protocol tailored to regional characteristics and centralized all MPOX notifications within a single information system [16].

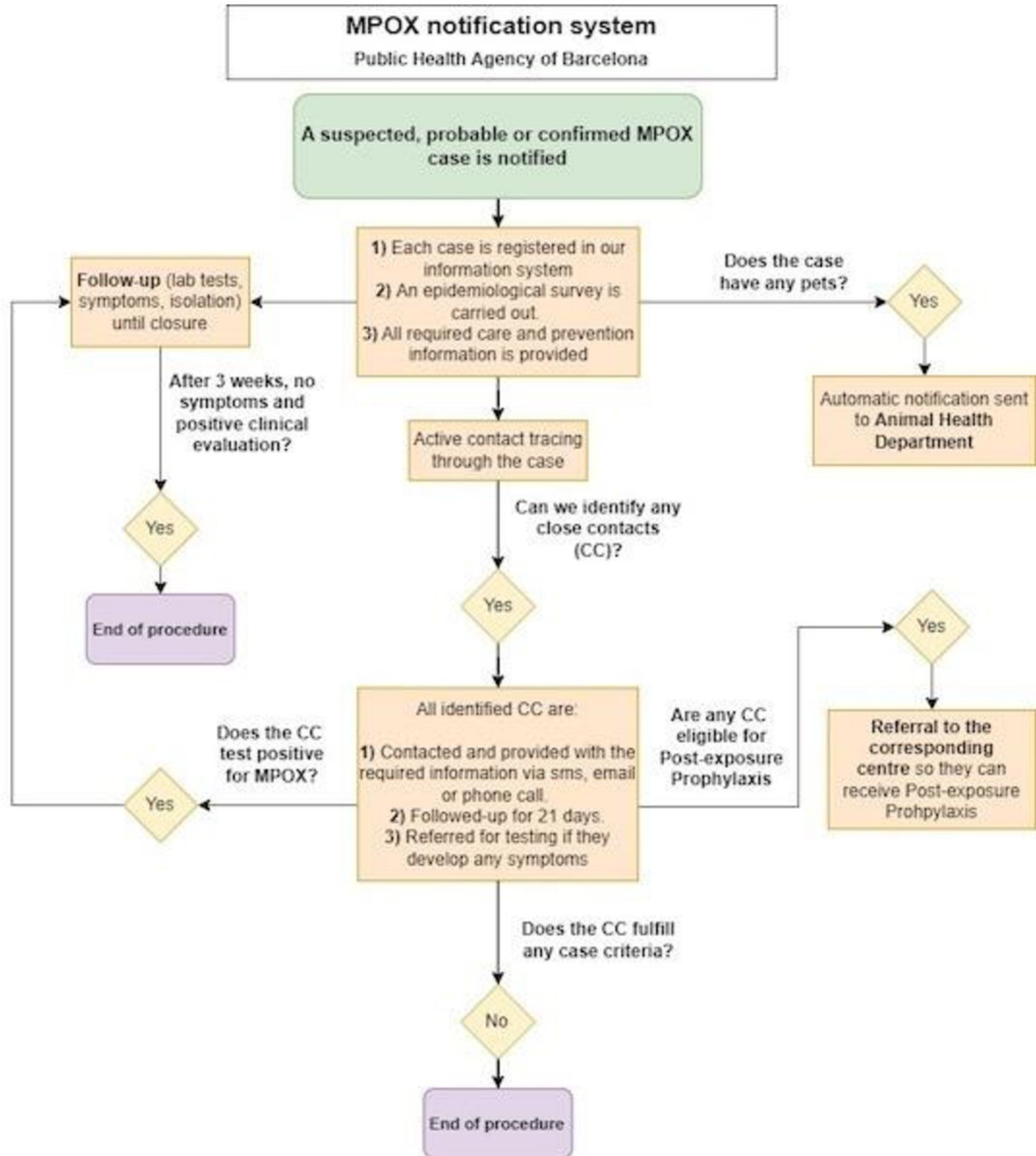

**Fig 1. Barcelona Public Health Agency 2022 MPOX notification system.** Source: Authors, according to guidelines from the Spanish Ministry of Health (14).

### Laboratory testing

In accordance with national and regional protocols from Spain [14] and Catalonia [15], during May 2022, all laboratory samples for MPOX testing were sent to Madrid and processed at the National Microbiology Center. Testing was decentralized from 31 July, by which time hospitals had validated over a hundred samples [15, 16]. Samples for PCR testing were collected from

vesicular liquid, exudate, or scab from the skin lesion, or from pharyngeal, oral, or anal mucosa if skin lesions were absent. In severe cases, a comprehensive microbiological study was performed, including genomic sequencing, and additional respiratory or neurological samples were collected depending on the site of complications [15, 16].

## Information sources

Each reported case in Barcelona was entered into a compulsory reporting platform by the healthcare worker suspecting the disease. CP and MG served as the designated epidemiologists responsible for the surveillance and with access to the complete dataset. To ensure patient confidentiality throughout the study, DP received only anonymized data for conducting the manuscript analyses, available at DOI: 10.5281/zenodo.13284254.

## Statistical analysis

The cumulative incidence per 1000 inhabitants, with 95% confidence intervals, was calculated based on the confirmed cases who were residents of the city, for both the general population and stratified by gender, using the city's 2022 population data. Absolute and relative frequencies were computed to identify the characteristics of confirmed cases and those requiring hospitalization. Explanatory variables included sociodemographic variables, such as age, sex, country of origin or place of residence, clinical symptoms and complications, and epidemiological variables, such as sex partners, recent travel in the last 21 days, attendance at mass events, contact with animals, previous smallpox vaccination, and contact tracing implementation.

Weekly epidemiological curves were plotted for the entire outbreak (May 2022- May 2023). Confirmed cases were georeferenced and assigned to their basic healthcare areas (BHA) within the city, which consisted of 65 territorial divisions based on the nearest primary care center, for geographical analysis. To assess differences in key populations, a stratified analysis was carried out based on the gender of their sexual partners. A GBMSM variable was created for confirmed cases of men reporting having sex with men or other genders besides women, compared to women or men who reported having sex only with women. Men who abstained from sex with both men and women were excluded from the analyses. Crude and adjusted multivariate analyses using Poisson regression models with robust variance were performed and are presented in a forest plot. The final model included all significant variables with adjustment for age and country of origin. Some categories were dichotomized or collapsed for better analyses, while significant clinical characteristics were excluded from the final models.

Data management and figure design were performed using Microsoft Office tools. Statistical analyses were carried out using the software STATA (c) v15, and geographical analyses were carried out using the software QGIS version 3.4.8-Madeira.

## Ethical concerns

This outbreak report is exempt from the requirement of approval by an ethics committee because it was conducted under the authority of Article 5.b. of the Catalonian Health Department Decree 203/2015, dated 15 September. This decree led to the establishment of the Epidemiological Surveillance Network and regulated the reporting systems for notifiable diseases and epidemic outbreaks. The study participants did not provide informed consent due to the regulatory framework governing of reporting systems for notifiable diseases and epidemic outbreaks. Five cases (0.3% of the sample) involved individuals aged 15–17 years, who required verbal consent from the parents or legal guardians.

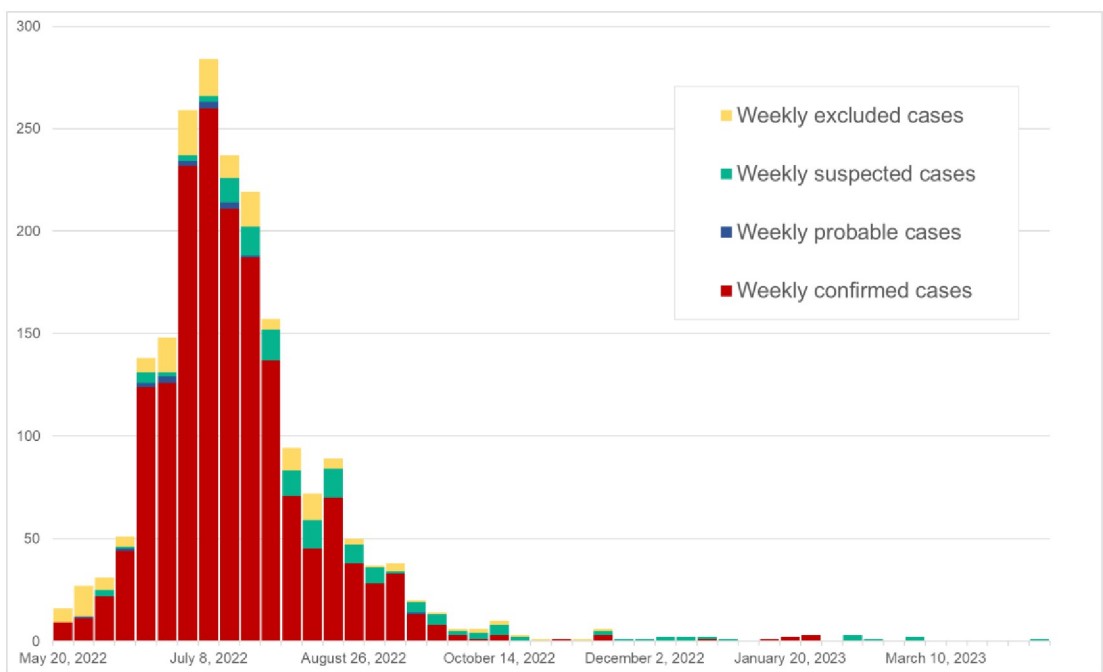

**Fig 2. Epicurve of the weekly MPOX notified cases in the city of Barcelona from May 20, 2022, to May 20, 2023.** Source: The compulsory MPOX notifications, collected by the authors from the Barcelona Public Health Agency (ASPB) from the Catalonia Epidemiological Surveillance Network.

## Results

As of May 20, 2023, a total of 2037 cases were reported among city residents in Barcelona, with a peak in July 2022. Of these, 82.6% (n = 1,684) were confirmed as cases, while 8.7% (n = 178) were excluded (Fig 2). The proportion of excluded cases decreased steadily over time.

Confirmed cases were predominantly male (98.2%), with 88.9% aged between 20 to 49 years. Men were older than women (median of 37 vs 33 years, p<0.001). Some 41.7% of cases were born in Spain, while 24.1% were from Latin American countries. Among the confirmed cases, 84.5% of men reported having sex with partners of the same gender, whereas 70.9% of women reported having sex only with partners of the opposite gender. About one-third of the cases had a prior HIV diagnosis and 10.6% had received a smallpox vaccination as a child. The cumulative incidence in the general population was 1.03 [95% CI 1.0–1.06] per 1000 inhabitants. It was 2.13 [2.01–2.17] per 1000 in men and 0.04 [0.03–0.05] per 1000 in women (Table 1).

The most common symptoms were the characteristic lesions, present in 82.9% of cases, with 48.1% having anal or genital lesions. General symptoms were reported in approximately 64.8% of cases, with fever being the most prevalent. Complications were described in 4.1% of the cases, and 1.6% required hospitalization (Table 2).

### Description of cases requiring in-hospital care

39.1% (n = 27) of the confirmed cases with any type of complication, required hospitalization. All hospitalized cases were men, with most of them presenting general symptoms (92.3%), with anal or genital lesions being the most common skin-related symptoms. None of the cases

**Table 1. Sociodemographic and epidemiological characteristics by sex, of the confirmed MPOX cases in Barcelona (May 2022-May 2023).**

| | Women (n = 31)<br>No. (%) | Men (n = 1653)<br>No. (%) | Total (n = 1684)<br>No. (%) |
|---|---|---|---|
| *Age (years)* | | | |
| *Under 20* | 2 (6.5%) | 11 (0.5%) | 13 (0.8%) |
| *20–29* | 10 (32.3%) | 314 (19.0%) | 324 (19.2%) |
| *30–39* | 6 (19.4%) | 675 (40.8%) | 681 (40.4%) |
| *40–49* | 9 (29.0%) | 483 (29.2%) | 492 (29.2%) |
| *50–59* | 2 (6.5%) | 139 (8.4%) | 141 (8.4%) |
| *60–69* | 1 (3.2%) | 25 (1.5%) | 26 (1.5%) |
| *>70* | 1 (3.2%) | 5 (0.3%) | 6 (0.4%) |
| *Missing values* | 0 | 1 (0.1%) | 1 (0.1%) |
| Case origin | | | |
| *Spain* | 12 (38.7%) | 692 (41.9%) | 704 (41.8%) |
| *Europe* | 1 (3.2%) | 273 (16.5%) | 274 (16.3%) |
| *America* (a) | 8 (25.8%) | 404 (24.4%) | 412 (24.5%) |
| *Asia* | 2 (6.5%) | 36 (2.2%) | 38 (2.3%) |
| *Africa* | 0 | 23 (1.4%) | 23 (1.4%) |
| *Oceania* | 0 | 2 (0.1%) | 2 (0.1%) |
| *Missing values* | 8 (25.8%) | 223 (13.5%) | 231 (13.7%) |
| Gender of sex partners | | | |
| *Same or other* | 2 (6.45%) | 1426 (86.3%) | 1428 (84.8%) |
| *Only opposite* | 22 (70.9%) | 30 (1.8%) | 52 (3.1%) |
| *Neither* | 7 (22.6%) | 193 (11.6%) | 200 (11.9%) |
| *Missing values* | 0 | 4 (0.2%) | 4 (0.2%) |
| Center of diagnosis | | | |
| *Primary care center* | 12 (38.7%) | 491 (29.7%) | 503 (29.9%) |
| *Hospital* | 13 (41.9%) | 800 (48.4%) | 800 (48.3%) |
| *STI specific center* | 6 (19.4%) | 343 (20.8%) | 343 (20.7%) |
| *Missing values* | 0 | 19 (1.2%) | 19 (1.1%) |
| Previous HIV diagnosis | 1 (3.2%) | 578 (34.9%) | 579 (34.4%) |
| Other causes of immunosuppression (not described) | 0 | 15 (0.9%) | 15 (0.9%) |
| Smallpox vaccine as child | 4 (12.9%) | 173 (10.5%) | 177 (10.5%) |
| MPOX vaccine during the outbreak | 0 | 22 (1.3%) | 22 (1.3%) |
| Close contact of a case | 7 (22.6%) | 424 (25.7%) | 431 (25.6%) |
| Travel in the previous 21 days | 7 (22.6%) | 186 (11.2%) | 193 (11.5%) |
| Attendance at a mass event | 5 (16.1%) | 429 (25.9%) | 434 (25.8%) |
| Lives with a pet | 10 (32.3%) | 330 (19.9%) | 340 (20.2%) |
| Probable route of transmission | | | |
| *Sexual* | 13 (41.9%) | 1390 (84.1%) | 1403 (83.3%) |
| *Person-to-person* (b) | 4 (12.9%) | 37 (2.2%) | 41 (2.4%) |
| *Other* (c) | 0 | 4 (0.2%) | 4 (0.2%) |
| *Unknown or missing* | 14 (45.2%) | 222 (13.4%) | 236 (13.9%) |
| Contact tracing | 9 (29.0%) | 129 (7.8%) | 139 (8.2%) |

(*Continued*)

**Table 1.** (Continued)

|  | Women (n = 31)<br>No. (%) | Men (n = 1653)<br>No. (%) | Total (n = 1684)<br>No. (%) |
|---|---|---|---|
| Cumulative incidence (95% CI) (d) | 0.02 (0.03–0.05) | 2.07 (2.01–2.17) | 1.03 (1.00–1.06) |

[a]Of the 412 American cases, only seven cases were originally from the USA and Canada, so only Latin American countries are included (24.1% vs 24.5% of the confirmed cases) in the Results section.

[b]Person-to-person, excluding sexual contact, mother-to-child or associated with the health care profession. Person-to-person includes face-to-face contact, such as talking or breathing close to one another, which can generate droplets or short-range aerosols; skin-to-skin; mouth-to-mouth; or mouth-to-skin contact.

[c]Other mechanisms also exclude blood transfusion, animal transmission, mother-to-child or associated with the health care profession, and unknown mechanism of transmission.

[d]Per 1000 inhabitants.

**Table 2. Clinical characteristics and complications of the confirmed MPOX cases, by sex.** Barcelona, 5/2022-5/2023.

|  | Women (n = 31) | Men (n = 1653) | Total (n = 1684) |
|---|---|---|---|
| General symptoms | 17 (54.8%) | 1074 (64.9%) | 1091 (64.8%) |
| *Fever* | 11 (35.5%) | 816 (49.4%) | 827 (49.1%) |
| *Asthenia* | 7 (22.6%) | 463 (28.0%) | 470 (27.9%) |
| *Odynophagia* | 11 (35.5%) | 327 (19.8%) | 338 (20.1%) |
| *Headache* | 9 (29.0%) | 288 (17.4%) | 297 (17.6%) |
| *Muscle pain* | 7 (22.6%) | 217 (13.1%) | 224 (13.3%) |
| Lymphadenopathies | 8 (25.8%) | 737 (44.6%) | 745 (44.2%) |
| *Local* | 7 (22.6%) | 626 (37.9%) | 633 (37.6%) |
| *General* | 2 (6.5%) | 216 (13.1%) | 218 (12.9%) |
| Exanthemas | 23 (74.2%) | 1372 (83.0%) | 1395 (82.4%) |
| *Anal genital* | 11 (35.5%) | 799 (48.3%) | 810 (48.1%) |
| *Oral-buccal* | 7 (22.6%) | 486 (29.4%) | 493 (29.3%) |
| *Other localizations* | 16 (51.6%) | 622 (37.6%) | 638 (37.9%) |
| Type of exanthema |  |  |  |
| *Pustular* | 5 (16.1%) | 250 (15.1%) | 255 (15.1%) |
| *Vesicular* | 7 (22.6%) | 245 (14.8%) | 252 (14.9%) |
| *Maculopapular* | 5 (16.1%) | 179 (10.8%) | 184 (10.9%) |
| *Umbilical* | 5 (16.1%) | 161 (9.7%) | 166 (9.9%) |
| *Scab-like* | 3 (9.7%) | 95 (5.8%) | 98 (5.8%) |
| *Hemorrhagic* | 0 | 4 (0.2%) | 4 (0.2%) |
| Complications | 1 (3.2%) | 68 (4.1%) | 69 (4.1%) |
| Hospitalization | 0 | 27 (1.6%) | 27 (1.6%) |
| Intensive care need | 0 | 0 | 0 |
| Bacterial infection | 1 (3.2%) | 13 (0.8%) | 14 (0.8%) |
| *Anal genital* | 1 (3.2%) | 5 (0.3%) | 6 (0.4%) |
| *Oral-buccal* | 0 | 4 (0.2%) | 4 (0.2%) |
| *Other localization* | 0 | 3 (0.2%) | 3 (0.2%) |
| Corneal infection | 0 | 7 (0.4%) | 7 (0.4%) |
| Bronchopneumonia | 0 | 1 (0.1%) | 1 (0.1%) |
| Deaths | 0 | 0 | 0 |

**Table 3. Description of 27 cases requiring hospitalization in Barcelona.** MPOX outbreak 5/2022-5/2023.

|  | Cases requiring hospitalization. (n = 27) |
|---|---|
| Age | 33.5 (27–43) (a) |
| Days of hospitalization | 3 (2–7.5) (a) |
| Male sex assigned at birth | 27 (100%) |
| Gender of sexual partners |  |
| *Same or other* | 23 (85.2%) |
| *Only opposite* | 2 (7.4%) |
| *Neither* (b) | 2 (7.4%) |
| Previous HIV diagnosis | 15 (55.6%) |
| Smallpox vaccine as a child | 4 (14.8%) |
| Close contact with a case | 10 (37%) |
| General symptoms | 25 (92.3%) |
| *Fever* | 23 (85.2%) |
| *Asthenia* | 12 (44.4%) |
| *Headache* | 9 (33.3%) |
| *Odynophagia* | 6 (22.2%) |
| *Muscle pain* | 4 (14.8%) |
| Lymphadenopathies | 16 (59.3%) |
| *Local* | 14 (51.9%) |
| *General* | 5 (18.5%) |
| Exanthema | 25 (92.6%) |
| *Anal or genital* | 18 (66.7%) |
| *Oral or buccal* | 13 (48.2%) |
| *Other localization* | 16 (59.3%) |
| Type of exanthema |  |
| *Pustular* | 8 (29.6%) |
| *Maculopapular* | 3 (11.1%) |
| *Vesicular* | 5 (18.5%) |
| *Umbilical* | 5 (18.5%) |
| *Scab-like* | 2 (7.4%) |
| *Hemorrhagic* | 1 (3.7%) |
| Bacterial infection (c) | 1 (3.7%) |
| Corneal infection | 2 (7.4%) |

(a) IQR: Interquartile range

(b) *Neither* includes two men who reported not having sex with women or men, with mechanism of transmission not related to sex.

(c) Bacterial infection in an oral-buccal exanthema, caused by *S. aureus*.

required intensive care unit management, and there were no deaths. Additionally, none of the hospitalized cases had received a smallpox vaccination during the current outbreak (Table 3).

## Key populations analyses

The highest number of cases and the most severe cases occurred among GBMSM. Geolocation data revealed that most of the cases lived around the city's main gay venues (Fig 3). A comparison between GBMSM and other cases revealed a positive association with being previously

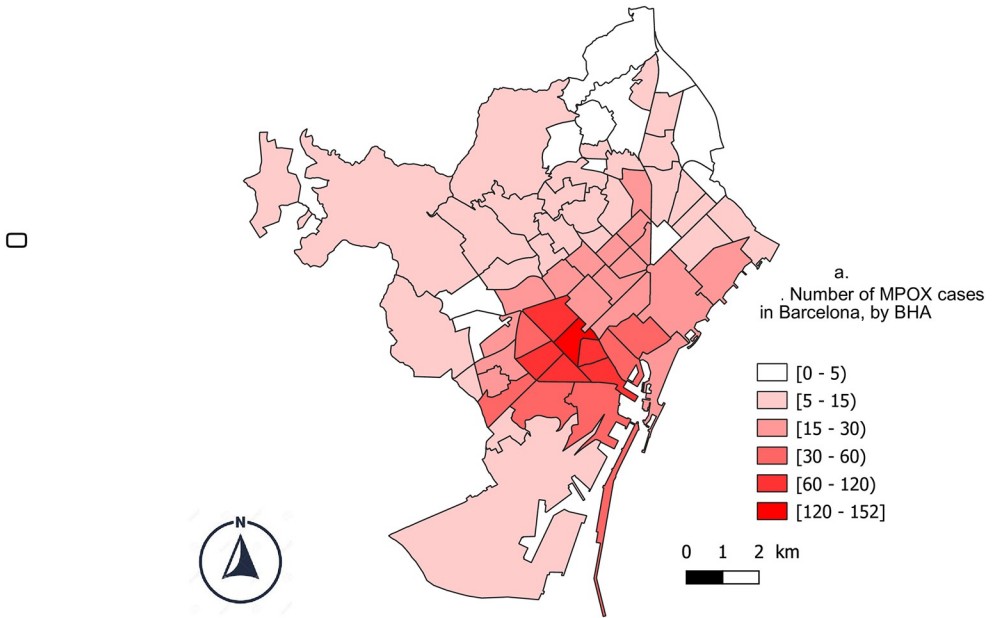

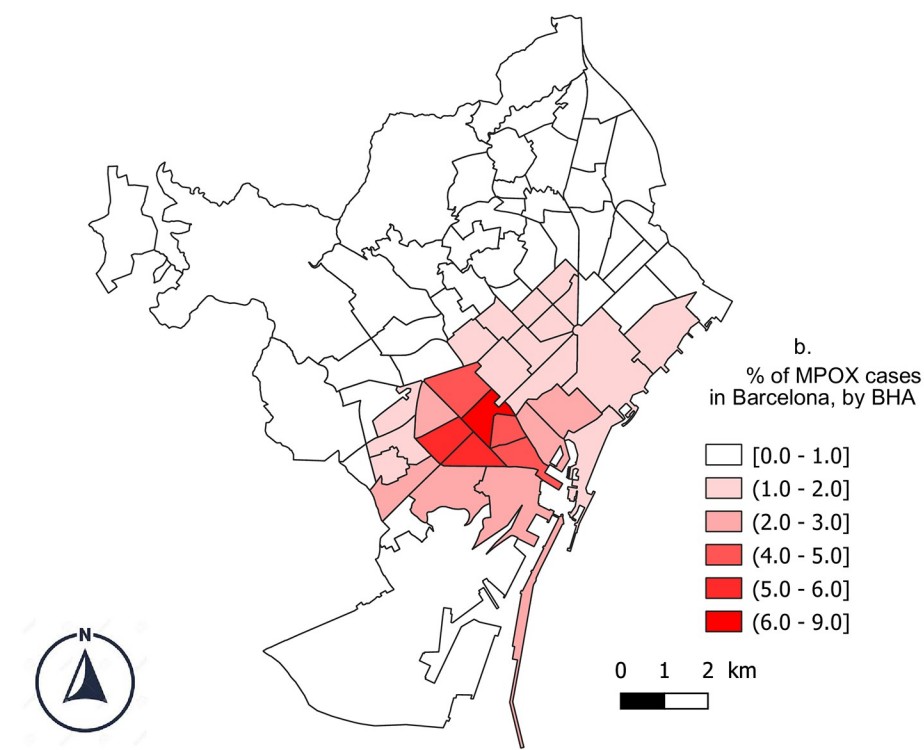

**Fig 3. Geolocation of confirmed cases during the first MPOX outbreak in Barcelona (5/2022-5/2023). Fig 3a** shows the total number of cases by BHA[(a)] while **Fig 3b** presents the percentage of cases by BHA residents. Maps were created with QGIS Geographic Information System version 3.4.8-Madeira, and all content is licensed under the Creative Commons Attribution–Share Alike 3.0 license (CC BY–SA), available at (https://qgis.org). In addition, geographical materials used for creating the map (e.g., shape file) were supported by Carto BCN from the Open data BCN project,

Barcelona City Council and MPOX data from the Epidemiology service registers. The authors specify that this figure is licensed under CC BY 4.0. (a) BHA: Basic Health Area.

diagnosed with HIV (APR: 1.07 CI95%: 1.04–1.11), attending public sexual venues (AOR: 1.06; 1.01–1.09), and not providing contact tracing information (AOR: 1.14; 1.06–1.22) (Fig 4).

## Prevention and control measures

Following the confirmation of the first case, prevention and control measures were implemented, including vaccination and community prevention strategies targeting cases, contacts and pets (Fig 1). Printed infographics, developed in collaboration with clinicians and local organizations, were posted at the entrances of gay saunas and sex clubs in the areas with the highest incidence. These messages advised individuals with MPOX lesions against entering these venues, highlighted the high incidence among GBMSM and provided brief recommendations on symptoms and management of suspected cases. A final message was issued to alert the public about cases with painful rectal lesions, promoting condom use with unknown partners, and stressing the importance of not sharing sex toys or douching material (S1 Appendix: Printed infographics). More detailed infographics were also published on social media (on the Instagram account of the rapid test program of the service, @provesrapides_aspb). These infographics addressed the need for stigma-free healthcare for key populations, risk reduction strategies related to the number of sexual partners, the use of condoms to prevent rectal lesions, and the impact of attending summer festivals for GBMSM, among other topics (S2 Appendix: Digital campaign).

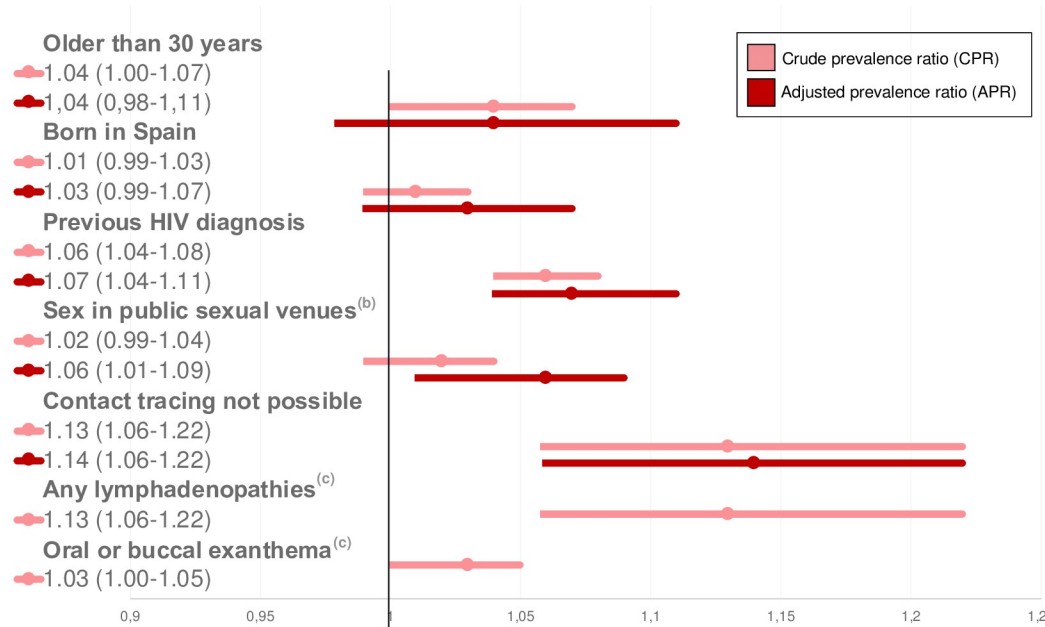

**Fig 4. Forest plot of the crude and adjusted prevalence ratios (with confidence interval of 95%) of the characteristics of the confirmed cases in GBMSM[a] in the first MPOX outbreak in Barcelona (5/2022-5/2023).** **(a)** GBMSM: gay, bisexual and other men who have sex with men. The variable was compared with cases occurring in women or men who only has sex with women. Those men who reported not having sex with men or women were excluded from the analyses. **(b)** Sex venues included saunas, sex clubs, parties, or cruising spots. **(c)** Clinical characteristics were excluded from final models due to the absence of an adjusted association or their inclusion as adjusting variables.

## Second outbreak

On August 17, 2023, a new MPOX outbreak was reported, with six confirmed cases. By the time of submission of this manuscript (December 2023), a total of 40 new MPOX cases have been confirmed in Barcelona, all among GBMSM, and of whom 37.5% had HIV. According to the available information, 27.5% had been vaccinated during the previous outbreak and 12.5% had received vaccination as a child. Five cases required hospitalization. Contact tracing was successfully completed for 90% of the cases.

## Discussion

The 2022–2023 MPOX outbreak was the largest recorded outside Africa, with Spain being one of the most severely affected countries, reporting 158.9 cases per a million inhabitants by May 2023 [17]. The city of Barcelona was among the hardest-hit cities globally, with 1,030 cases per million inhabitants. The rapid spread of this outbreak coincided with the lifting of the COVID-19 pandemic restrictions, leading to a substantial increase in global movement, especially among a highly intertwined international network of sexually active GBMSM [18]. Furthermore, the city of Barcelona may have specific social and cultural characteristics that influence tourism choices and the nightlife industry [19], which could act as a hotspot for GBMSM, thereby increasing the risk of transmission between local residents and travelers.

Sexual transmission was the most common route of infection among confirmed cases, with GBMSM accounting for the majority of both cases and severe cases. This group was acknowledged as a vulnerable population in reports of MPOX outbreaks in other regions [17, 20–24]. Likewise, GBMSM were closely associated with previous STI outbreaks in the city of Barcelona [25, 26]. The geolocation of the cases was strongly correlated with local gay venues and the previous spatial visualization of HIV diagnoses in Catalonia [27], reinforcing the association of this group with the outbreak. This information could be used for tracking the spread of diseases, identifying high transmission areas, and planning interventions to control future outbreaks [28].

STIs pose a global threat, especially–but not exclusively–to non-monogamous populations with multiple sexual partners, such as some GBMSM. In our sample, non-GBMSM were significantly younger. An upward trend in STI rates was reported among the youth in most high-income countries [29], underscoring the need for further analyses and interventions. The disease was generally mild and self-limiting, with an absence of local lethal cases. Proctitis, which was reported as one of the most severe complications early in the outbreak, was also identified in different regions [30, 31]. Preventing proctitis was a key message of our printed prevention campaigns for saunas and sex clubs (S1 Appendix), providing an opportunity to reinforce the promotion of condom usage, particularly in scenarios involving sexual encounters with unknown or untraceable partners [17, 21–23].

We encountered major challenges in case reporting and surveillance (CRS), especially in contact tracing, with only 8.2% of the cases correctly traced. Tracing was especially difficult in the GBMSM group, which aligns with reports of other MPOX outbreaks [17, 21–23]. This rate was lower compared to contact tracing for HIV cases in the same year (40.2%) [32], a discrepancy that may be influenced by the rapid increase of cases and the broader pandemic context, although further research is needed. The use of geosocial networking (GSN) apps to locate sexual partners has been a common source of exposure in Spain [1] and may translate into a major barrier to partner notification. Consequently, health surveillance systems should consider integrating GSN apps into future contact tracing strategies [33]. The presence of our rapid test program on these apps [34] helped address queries, facilitate linkage to vaccination

programs, and provide trustworthy and non-stigmatized information [24]. Nevertheless, further interventions are welcome.

The MPOX-CRS varies across countries. In Spain, the case definition included acceptance of *Orthopoxvirus*-specific PCR as a diagnostic criterion and the inclusion of arthralgia as a clinical symptom. Sequencing was not required to confirm diagnosis [35]. Standardization strengthens public health preparedness and response and creates unified communication [35–37]. Giarducci also states that the current MPOX case definitions provided by the WHO and ECDC can result in missed cases occurring in heterosexual patients with a characteristic vesicular-pustular rash but no travel history or contact with confirmed cases, as observed in the present study. In our records, some variables were not documented, such as the use of sexualized drugs, diagnosis of an STI coinfection, HIV pre-exposure prophylaxis (PrEP) usage, or whether a person was transgender or not, creating a limitation for those analyses.

Following the lifting of COVID-19-related lockdowns, some regions observed a significant increase in sexual practices that can facilitate STI transmission among GBMSM, such as group sex, encounters with unknown partners, ineffective use of barrier methods, and chemsex [38]. Furthermore, epidemiological surveillance systems had to address a new public health concern amidst an ongoing pandemic, with the MPOX outbreak coinciding with the onset of the seventh COVID-19 wave, thus hindering the effective integration of lessons learned by healthcare professionals [18, 24, 39]. Those lessons include the need for an automatized system able to developing interventions based on real-time case classification, collecting significant data for the decision-making process and implementing effective contact tracing. Moreover, developing strategies to support the tracers' activities and engaging diverse community agents would enhance the effectiveness of surveillance systems [36].

Contact tracing can significantly increase the time spent in investigating outbreaks, overloading surveillance workers. Some authors have designed a tracing tool that iteratively chooses an individual from the set to query, based on 21 variables, so that tracers can select the persons allowing retrieval of a larger number of contacts, thus focusing efforts more efficiently [40]. Such tools can be useful to gather the highest number of contacts and improve the results of epidemiological tasks in times of public health emergencies of international concern, as was the case of this outbreak. Other strategies include intersectoral collaboration, robust diagnostic tools, clinical management plans, firm prevention plans, capacity building, sociocultural and emotional communication strategies addressing stigma and discrimination against groups with greater vulnerability, and ensuring equitable access to treatments and vaccines [24, 37].

In various regions, sexual health clinicians played a front-line role in responding to the outbreak given the affected patient population, highlighting the need for reinforced services [22]. The end of this MPOX outbreak may depend on the expansion of the vaccine campaign or the success of non-pharmacological and community-based measures. However, broader seroprevalence studies are needed to determine their impact on reducing the diagnosis of mild or asymptomatic cases and to assess whether saturation has been reached within the exposed population. Additionally, reinfections and new cases have been notified in Barcelona, hypothetically due to the human movements for the 2023 summer season and festivals, especially those aimed toward GBMSM. The lower incidence and lessons learned have helped us improve the contact tracing, but we may need to standardize the care for a new regular STI.

There has been considerable discussion about the reasons for the resurgence of MPOX cases, with the prevailing theory being waning immunity. Other factors involved in the resurgence or even boost in the number of cases may be climate change, deforestation of the rain forest, geopolitical conflicts, and a highly mobile population [41]. Smallpox vaccination has been proven to reduce the risk of severe MPOX disease by up to 85% [4]. Nevertheless, following its eradication in 1980 and considering the risk of disseminated vaccinia as a severe

complication of the smallpox vaccine in HIV-positive patients during the emergence of AIDS [42], routine vaccination was no longer indicated. This has led to four decades passing without any *Orthopoxvirus* vaccination program worldwide [3, 4]. Our data were unable to capture the progress of vaccination comprehensively, and we found no significant differences in vaccination rates between the total sample and persons who were hospitalized. Therefore, further research is needed in this field to assess the need for new surveillance and vaccination programs, while also examining motivations for and against receiving vaccination [43].

During the past year, MPOX transformed from a nearly-forgotten disease into a sexually transmitted disease of international concern. Surveillance and research on new POX viruses, such as the Akhmeta virus or Alaskapox virus, also require further discussion. This discussion should include the identification of wild species that harbor these viruses in various areas of Africa, and a more precise definition of the clinical spectrum and severity of the disease, including asymptomatic carriage and risk factors for its acquisition. Moreover, an improved description of the outbreak patterns in terms of size and duration, and measurement of the risk of transmission associated with different types of contact with clinical cases should be investigated [41]. By incorporating the One Health perspective, surveillance teams can stay regularly up-to-date on emerging diseases and be prepared for future emergencies [18, 24].

## Conclusions

The 2022 MPOX outbreak has posed a major challenge to surveillance and sexual health services. The city of Barcelona has experienced one of the most substantial impacts worldwide, with characteristics that could potentially facilitate the occurrence and spread of future emergencies. The GBMSM population has once again been the most strongly affected group, highlighting the need for further interventions in sexual health promotion and surveillance. A One Health perspective is essential for preparedness for emerging diseases.

## Supporting information

**S1 Appendix. Printed infographics.** Posters displayed at local saunas and sex clubs during August 2022.
(DOCX)

**S2 Appendix. Digital campaign.** Infographics published in the Instagram account @provesrapides_aspb.
(DOCX)

## Acknowledgments

We wish to thank the nurses, physicians and community health care workers from the epidemiology department involved in the surveillance, report and contact tracing of this outbreak. We also acknowledge the community entities who became involved in prevention strategies early in the outbreak and the sexual health care workers who played a front-line role in the emergency.

## Author Contributions

**Conceptualization:** David Palma, Anna de Andrés, Paula Santiá, Patricia García de Olalla, Cristina Rius.

**Data curation:** David Palma, Montserrat Guillaumes, Anna de Andrés, Jesús Ospina, Paula Santiá.

**Formal analysis:** David Palma, Paula Santiá.

**Funding acquisition:** Cristina Rius.

**Investigation:** David Palma, Montserrat Guillaumes, Carles Pericas, Anna de Andrés, Jesús Ospina, Paula Santiá, Patricia García de Olalla.

**Methodology:** David Palma, Paula Santiá, Patricia García de Olalla.

**Project administration:** Anna de Andrés, Cristina Rius.

**Resources:** Cristina Rius.

**Software:** Raquel Prieto.

**Supervision:** Patricia García de Olalla, Cristina Rius.

**Validation:** David Palma, Montserrat Guillaumes, Anna de Andrés, Cristina Rius.

**Visualization:** David Palma, Carles Pericas, Raquel Prieto.

**Writing – original draft:** David Palma, Carles Pericas, Laia Álvarez-Bruned.

**Writing – review & editing:** David Palma, Laia Álvarez-Bruned, Jesús Ospina, Cristina Rius.

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
