## [Decision Letter · Decision Letter 0]

26 Jun 2024

PONE-D-23-38786A new STI in the city: MPOX in Barcelona. First outbreak (5/2022-5/2023) and subsequent resurgencePLOS ONE

Dear Dr. Palma,

Thank you for submitting your manuscript to PLOS ONE. After careful consideration, we feel that it has merit but does not fully meet PLOS ONE’s publication criteria as it currently stands. Therefore, we invite you to submit a revised version of the manuscript that addresses the points raised during the review process.

We look forward to receiving your revised manuscript.

Kind regards,

Milad Zandi, Ph.D.

Academic Editor

PLOS ONE

Journal Requirements:

5. We note that Figure 3 in your submission contain map/satellite images which may be copyrighted. All PLOS content is published under the Creative Commons Attribution License (CC BY 4.0), which means that the manuscript, images, and Supporting Information files will be freely available online, and any third party is permitted to access, download, copy, distribute, and use these materials in any way, even commercially, with proper attribution. For these reasons, we cannot publish previously copyrighted maps or satellite images created using proprietary data, such as Google software (Google Maps, Street View, and Earth). For more information, see our copyright guidelines: http://journals.plos.org/plosone/s/licenses-and-copyright.

a. You may seek permission from the original copyright holder of Figure 3  to publish the content specifically under the CC BY 4.0 license.  

Reviewers' comments:

Reviewer's Responses to Questions

**Comments to the Author**

1. Is the manuscript technically sound, and do the data support the conclusions?

Reviewer #1: Yes

Reviewer #2: No

2. Has the statistical analysis been performed appropriately and rigorously? 

Reviewer #1: Yes

Reviewer #2: No

3. Have the authors made all data underlying the findings in their manuscript fully available?

Reviewer #1: Yes

Reviewer #2: No

4. Is the manuscript presented in an intelligible fashion and written in standard English?

Reviewer #1: Yes

Reviewer #2: No

5. Review Comments to the Author

Reviewer #1: This publication on an emerging infectious disease is quite important to its global preventive measures. The experience of the authors in the identification of the most susceptible group that are prone to Mpox will inform public health intervention by health leaders globally.

Reviewer #2: The manuscript addresses an interesting topic but has several issues that need to be addressed:

1. The structure does not follow a typical descriptive study format. Key sections like inclusion and exclusion criteria for notified cases, as well as statistical analysis, are missing or inadequately described.

2. Essential descriptive data, such as the proportion of immunosuppression status, prior vaccination status for orthopoxvirus, and comprehensive clinical and laboratory findings, are not well-presented in the results section. Figures, charts, and tables lack adequate text descriptions.

3. The authors label the study as the "first report" of MPOX in the region, despite existing research reporting MPOX in the same area (e.g., https://doi.org/10.1093/ofid/ofae105).

4. The abstract is missing a background section, and the original source of Figure 1 is not cited.

6. PLOS authors have the option to publish the peer review history of their article (what does this mean?). If published, this will include your full peer review and any attached files.

Reviewer #1: **Yes: **Ado Garba Abubakar

Reviewer #2: **Yes: **zahra heydarifard

---

## [Author Response · Author response to Decision Letter 0]

9 Aug 2024

1. Is the manuscript technically sound, and do the data support the conclusions? 

Reviewer #1: Yes 

Reviewer #2: No 

Methodology has been updated, with changes in the inclusion and exclusion criteria, the statistical analyses developed and the source of some of the presented images. 

2. Has the statistical analysis been performed appropriately and rigorously? 

Reviewer #1: Yes 

Reviewer #2: No 

We believe that our data has been extensively analyzed, however, we understand that some information could be improved, such as immunosuppression status, prior vaccination status for orthopoxvirus, and comprehensive clinical and laboratory findings. We have updated the results with the available information previously non presented. 

3. Have the authors made all data underlying the findings in their manuscript fully available? 

Reviewer #1: Yes 

Reviewer #2: No 

Anonymized data has been deposited in a public repository. 

4. Is the manuscript presented in an intelligible fashion and written in standard English? 

Reviewer #1: Yes 

Reviewer #2: No 

Although a native revision was made before the submission, a new one was conducted. Improvements in English language are marked in yellow highlight. 

5. Review Comments to the Author 

Reviewer #1: This publication on an emerging infectious disease is quite important to its global preventive measures. The experience of the authors in the identification of the most susceptible group that are prone to Mpox will inform public health intervention by health leaders globally. 

We appreciate the kind words, and hope that our results, as you comment, could help the develop of public health interventions globally. 

Reviewer #2: The manuscript addresses an interesting topic but has several issues that need to be addressed: 

1. The structure does not follow a typical descriptive study format. Key sections like inclusion and exclusion criteria for notified cases, as well as statistical analysis, are missing or inadequately described. 

We appreciate the comment provided and have made efforts to improve the outbreak manuscript format to follow a descriptive study. In that line, inclusion and exclusion criteria have been added as a single paragraph, while statistical analyses have been presented separately of source of information. 

2. Essential descriptive data, such as the proportion of immunosuppression status, prior vaccination status for orthopoxvirus, and comprehensive clinical and laboratory findings, are not well-presented in the results section. Figures, charts, and tables lack adequate text descriptions. 

Changes have been made to all the commented issues with the available information. Also, the figures, charts and tables descriptions have been improved. However, laboratory findings were not presented, due to the high amount of missing data, with individuals who received both a generic POX and specific MPOX testing, while others who did not have information. We believe, reviewing similar literature, that the absence of information does not affect the outbreak's description. 

3. The authors label the study as the "first report" of MPOX in the region, despite existing research reporting MPOX in the same area (e.g., https://doi.org/10.1093/ofid/ofae105). 

This is an important debate because we can confidently state that our complete data is not fully represented in the results presented in this article. For instance, a single city has reported more diagnoses than the 52 notifying centers combined. While we acknowledge the contributions of other authors, their study reports 1,472 cases, whereas ours presents 1,684. 

The study by Ramirez-Olivencia et al., a descriptive analysis of a series of cases, emphasizes that the majority of cases were reported in Madrid, Catalonia, and Andalusia. In contrast, our analysis, which is an outbreak report, focuses specifically on a single city—Barcelona (part of Catalonia)—and includes time-specific measures and preventive actions. We observed that while two centers in Barcelona may have reported cases to the study, they likely did not account for the total number of cases from each center. Additionally, it is probable that most cases included in the study were reported from Badalona, a city close to Barcelona, where the Hospital Can Ruti is located, and where some of the study's authors practice. 

4. The abstract is missing a background section, and the original source of Figure 1 is not cited. 

Changes have been made in order to improve the quality of the abstract, as well as the original source of figure 1.

---

## [Decision Letter · Decision Letter 1]

7 Oct 2024

PONE-D-23-38786R1A new STI in the city: MPOX in Barcelona. First outbreak (5/2022-5/2023) and subsequent resurgencePLOS ONE

Dear Dr. Palma,

Thank you for submitting your manuscript to PLOS ONE. After careful consideration, we feel that it has merit but does not fully meet PLOS ONE’s publication criteria as it currently stands. Therefore, we invite you to submit a revised version of the manuscript that addresses the points raised during the review process.

We look forward to receiving your revised manuscript.

Kind regards,

Vikash Jaiswal, MD

Academic Editor

PLOS ONE

Journal Requirements:

Additional Editor Comments:

Dear Authors

Please revise the draft as per reviewer suggestions.

Reviewers' comments:

Reviewer's Responses to Questions

**Comments to the Author**

1. If the authors have adequately addressed your comments raised in a previous round of review and you feel that this manuscript is now acceptable for publication, you may indicate that here to bypass the “Comments to the Author” section, enter your conflict of interest statement in the “Confidential to Editor” section, and submit your "Accept" recommendation.

Reviewer #1: All comments have been addressed

Reviewer #2: All comments have been addressed

2. Is the manuscript technically sound, and do the data support the conclusions?

Reviewer #1: Yes

Reviewer #2: Yes

3. Has the statistical analysis been performed appropriately and rigorously? 

Reviewer #1: Yes

Reviewer #2: Yes

4. Have the authors made all data underlying the findings in their manuscript fully available?

Reviewer #1: Yes

Reviewer #2: Yes

5. Is the manuscript presented in an intelligible fashion and written in standard English?

Reviewer #1: Yes

Reviewer #2: No

6. Review Comments to the Author

Reviewer #1: The authors have adequately addressed my comments on statistical analysis in the revised version of the manuscript.

Reviewer #2: The manuscript needs to be thoroughly proofread for typos and language improvements. For instance, 'COVID-19' should be written in all caps throughout the text. All changes made by the authors should be highlighted in the manuscript.

7. PLOS authors have the option to publish the peer review history of their article (what does this mean?). If published, this will include your full peer review and any attached files.

Reviewer #1: **Yes: **Ado G Abubakar

Reviewer #2: **Yes: **Zahra Heydarifard

---

## [Author Response · Author response to Decision Letter 1]

23 Nov 2024

Thank you for your kind revisions.

---

## [Editor Report · Decision Letter 2]

28 Nov 2024

A new STI in the city: MPOX in Barcelona. First outbreak (5/2022-5/2023) and subsequent resurgence

PONE-D-23-38786R2

Dear Dr. Palma,

We’re pleased to inform you that your manuscript has been judged scientifically suitable for publication and will be formally accepted for publication once it meets all outstanding technical requirements.

Kind regards,

Vikash Jaiswal, MD

Academic Editor

PLOS ONE

Additional Editor Comments (optional):

Looks all comment has been resolved. 
---

## [Editor Report · Acceptance letter]

3 Jan 2025

PONE-D-23-38786R2 

PLOS ONE

Dear Dr. Palma, 

I'm pleased to inform you that your manuscript has been deemed suitable for publication in PLOS ONE. Congratulations! Your manuscript is now being handed over to our production team.

Kind regards, 

on behalf of

Dr. Vikash Jaiswal 

Academic Editor

PLOS ONE